# What Is the Evolutionary Fingerprint in Neutrophil Granulocytes?

**DOI:** 10.3390/ijms21124523

**Published:** 2020-06-25

**Authors:** Leonie Fingerhut, Gaby Dolz, Nicole de Buhr

**Affiliations:** 1Department of Physiological Chemistry, Department of Infectious Diseases, University of Veterinary Medicine Hannover, 30559 Hannover, Germany; leonie.fingerhut@tiho-hannover.de; 2Research Center for Emerging Infections and Zoonoses (RIZ), University of Veterinary Medicine Hannover, 30559 Hannover, Germany; 3Clinic for Horses, University of Veterinary Medicine Hannover, 30559 Hannover, Germany; 4Escuela de Medicina Veterinaria, Universidad Nacional, Heredia 40104, Costa Rica; gaby.dolz.wiedner@una.ac.cr

**Keywords:** neutrophil, species, granulocyte

## Abstract

Over the years of evolution, thousands of different animal species have evolved. All these species require an immune system to defend themselves against invading pathogens. Nevertheless, the immune systems of different species are obviously counteracting against the same pathogen with different efficiency. Therefore, the question arises if the process that was leading to the clades of vertebrates in the animal kingdom—namely mammals, birds, amphibians, reptiles, and fish—was also leading to different functions of immune cells. One cell type of the innate immune system that is transmigrating as first line of defense in infected tissue and counteracts against pathogens is the neutrophil granulocyte. During the host–pathogen interaction they can undergo phagocytosis, apoptosis, degranulation, and form neutrophil extracellular traps (NETs). In this review, we summarize a wide spectrum of information about neutrophils in humans and animals, with a focus on vertebrates. Special attention is kept on the development, morphology, composition, and functions of these cells, but also on dysfunctions and options for cell culture or storage.

## 1. Introduction

As the first line of response during immune reactions against invading pathogens, neutrophil granulocytes are an important blood cell type of animals. Neutrophils mature in the bone marrow, to then be released into the blood circulation, but may also be stored in a marginal pool on blood vessel walls [1]. From blood vessels, neutrophils can transmigrate through vascular endothelium into infected tissue. They find their way towards their target point guided by chemotactic agents. Once having reached the infected tissue, neutrophils can react with various defense strategies, including degranulation, phagocytosis, or release of neutrophil extracellular traps (NETs) [2]. As such, they combat either the pathogens with antimicrobial substances or engulf and digest them. Furthermore, they can entrap microbes in released extracellular chromatin structures decorated with histones and granule proteins. Nevertheless, these mechanisms also have a detrimental potential, which can lead to destruction of healthy surrounding tissue or even autoimmune reactions [3,4,5].

Already among the neutrophils of one species, numerous modifications are present, even in a healthy state. For instance, immature band neutrophils and mature segmented neutrophils can be detected inside the blood. Furthermore, an occurrence of various subtypes of neutrophil granulocytes is discussed [6,7]. Such a heterogeneous population could explain the two-sided role of neutrophils in diseases, as it was suggested for human asthma [8]. Consequently, it is evident to assume the evolutionary processes creating thousands of different animal species might likewise generate differences among the neutrophil granulocytes. A widely known example would be the occurrence of heterophils instead of neutrophils in birds, as well as in some mammalian and reptile species. Furthermore, besides morphologic differences, species variations in neutrophil function exist, as it has been recently summarized in the regard of NETs [9].

In this review, we summarize a wide spectrum of information about neutrophils in vertebrates with a focus on evolution, development, morphology, and function.

## 2. Evolution

To understand the evolutionary pathways finally leading to highly specialized cell forms like neutrophils, it might be constructive to investigate further blood cells or related cellular forms in animals without or with a less developed vascular system.

Phagocytic cells have been detected in all multicellular animals and their origin is proposed to be in protozoans already [10]. In basic invertebrates without a coelom, and thus without a vascular system, these freely moving cells possessing some blood cell characteristics are entitled as amebocytes, neoblasts, or interstitial cells [11]. Such cells which take up and digest particles were found in sponges already [10,11], while coelenterates were described to have cells which may act as phagocytes (reviewed in [11]). Meanwhile, freely moving cells of flatworms were compared to hemocytoblasts due to their ultrastructural appearance (reviewed in [11]). However, it remains to be clarified whether one of these cell types is homologous to hemocytes, the phagocytes of invertebrates found in their hemolymph [11]. However, it is proposed that these phagocytic cells, which had originally the sole function of food supply, correspond to mammalian phagocytic cells [10]. Comparisons of granule contents have opened a further discussion, as higher similarities between macrophages of vertebrates and phagocytes of invertebrates were found [12]. Nevertheless, for instance the granule enzyme myeloperoxidase was detected in the neutrophil granules of many vertebrate species, as well as in some invertebrates [13]. For instance, in freshwater snails high peroxidase values were detected, while the tested terrestrial snail lacked this enzyme [13].

In invertebrates with a secondary body cavity, a vascular system with four types of hemocytes can be found, of which one type is termed as granular hemocytes. These cells are typically packed with enzyme-filled granules. Furthermore, even neutrophilic, eosinophilic, and basophilic granulocytes may be distinguished in many invertebrates, just like in vertebrates. Another similarity was discovered regarding the function of those cells, as they contribute, among other things, to immune and metabolic processes [11]. As some deuterostomes, such as *Amphioxus*, possess granular hemocytes [14] but for instance echinoderms do not, it is suggested that the development of the blood cell types seen in vertebrates occurred during the early evolution of this taxon [11]. This would be in accordance to the monophyletic scheme of phylogenetic relationships of animals, in which vertebrates evolve from the deuterostomes [10] (Figure 1).

Despite these relationships, numerous similarities can also be seen between insect hemocytes and mammalian phagocytes. As such, insects represent a popular model for the innate immune system [15]. Even though the morphologic appearance of those cells is distinct, for instance the recognition mechanisms, phagocytosis, and killing of pathogens appear quite similar [15,16,17]. Especially the plasmatocytes and granular cells, two sub forms of insect hemocytes, resemble neutrophils. Both cell types manage to combat microbes by phagocytosis involving a respiratory burst, using alike receptors and antimicrobial peptides [15]. Interestingly, the evolution of antimicrobial peptides in Diptera (true flies) is discussed to depend on the presence of pathogens or factors of the environment [18].

Another approach to investigate the development of neutrophil functions is the closer examination of the postnatal development of neutrophils in different species. Of special interest are marsupials like the opossum, as their newborns are not equipped with a developed immune system. They are thus considered a model for investigations regarding the evolution of the mammalian immune system [19]. Opossums are born with only a low number of neutrophils, among these many immature forms. Postnatally, the number of neutrophils increases until they reach an about 40-fold increase after 15 days, and later on again decrease a bit. Additionally, the opossum neutrophils show a transition regarding their lobulation during the postnatal period. At four to five days of age, the immature granulocytes cease and for a period of 12 days the number of nuclear lobes increases [20]. Similar pictures of neutrophil development post-partum are seen in other marsupials, such as the quokka, eastern quoll, brushtail possum, southern brown bandicoot, and the tammar wallaby. In these species, immature myeloid cells dominate the white blood count at birth, with only rare appearances of neutrophil myelocytes. The neutrophils rapidly become the predominant leukocyte type in the kangaroo species, but later on decrease while lymphocyte numbers increase, reaching a 1:1 ratio at the age of a month. Finally, a differential leukocyte composition comparable to adults is reached by 80 days of life [21,22]. This development of blood cells is similar to the one occurring during gestation in eutherian animals. Hereby, the yolk sac is the first site of hematopoiesis in the embryo, along with the dorsal aorta. In the fetus it is changing to liver and spleen and ultimately to the bone marrow by the end of gestation [23,24]. In cats, the first appearance of neutrophils is already by 17 days post conception during the stage of yolk sac hematopoiesis [25]. In pigs and cattle the first neutrophils are derived from the fetal liver [25,26]. In humans, only neutrophil progenitors are present in the fetal liver, but no mature forms [27]. Instead, neutrophils first appear by 10–11 weeks of gestation in the human bone marrow [28], and mature forms are released starting about three weeks later [27]. The difference in marsupials is the emergence of the hematopoietic development stages in liver, spleen, and bone marrow only after birth during the maturation period inside the pouch [22].

As the postpartum development in marsupials is alike the prenatal one in eutherians, it naturally contrasts with the eutherians’ postnatal evolution. In horses, foals possess higher absolute neutrophil counts than adults, especially fillies [29]. Equally in species such as cows and pigs, neutrophils account for the main part of white blood cells postnatally. Nevertheless, in these animals a transition period follows for about two weeks, after which the characteristic adult lymphocytic hemogram develops [1,30]. In humans, the amount of neutrophils in the periphery greatly increases until it stabilizes after two to three days [31]. Alike in opossums, neonatal human neutrophils bear qualitative deficiencies [20,32]. These appear in form of impaired adhesion and decreased directed migration in human newborns [33,34]. Likewise, puppies are born with a higher amount of immature neutrophils, but these band neutrophils decrease to reference values within 7–10 days [1].

Overall, the influence of evolution on neutrophil nature and function in different species seems conceivable. Therefore, we will compare the nature of neutrophils from different species of the animal kingdom and the role of neutrophils in some diseases in the next parts.

## 3. Cell Number and Granulopoiesis

### 3.1. Cell Number

Neutrophils represent the predominant cell type in the blood of many, but not all mammals [35]. By contrast, in teleost fish less than 5% of the circulating leukocytes are neutrophils and the major part of neutrophils is stored in the kidney [36]. Depending on the species, lymphocytic and granulocytic hemograms can be distinguished (Table 1). With a proportion of more than 50% neutrophils of the total blood leukocytes, most carnivores belong to the granulocytic group, as well as horses. Granulocytes also dominate in human blood. In comparison, ruminants and laboratory rodents only have 20–30% neutrophils and are consequently examples for animals with a lymphocytic hemogram [35]. Other species belonging to the lymphocytic group are fishes and opossums. Pigs are allocated to the lymphocytic or the granulo-lymphocytic group, with up to even amounts of these two cell types [37,38]. In birds and reptiles, species with lymphocytic (e.g., turkey, chicken, snake) and those with heterophilic hemograms (e.g., grey parrots and budgerigars) exist [37,39]. Turtles can be allocated to both groups due to their wide range of 21–74% heterophils in total blood leukocytes. Furthermore, variations within species can be observed. For instance, turkeys of meat production lines have higher values of heterophils and erythrocytes than egg producing turkeys [40].

Nevertheless, cell counts also vary depending on factors such as health status and environmental influences. Leukocyte numbers in horses rise for example due to stress, inflammation, or acute to chronic infections and decrease in case of viral infections, endotoxins or acute peritonitis, pleuritis or colitis [41]. Hereby, especially the number of segmented neutrophils increases in inflammation, particularly if bacterial-purulent, following administration of corticosteroids or as an effect of adrenaline. Examples for stress-induced neutrophilia are the effects of transportation or strenuous exercise observed in horses, the latter causing an increase for 24 h after the end of exercise [42]. A stress leucogram is also frequently seen in pigs and cats. Such a response develops within two minutes due to a mobilization of leukocytes from the marginal into the circulating pool [43]. Parturition is another stressful factor influencing the white blood count, leading to neutrophilia for example in pigs and cattle, but reversing to normal within a day [1]. The magnitude of neutrophilia due to inflammation is less intense in cattle than in dogs, cats, or horses. This is due to a comparatively small storage pool in cattle. Therefore, an increase of neutrophils primarily originates from increased production instead of mobilization from the marginal pool, as it is seen in other species [44]. The values of segmented neutrophils drop in case of processes that go beyond the regenerative capacity of the bone marrow. A left shift occurs if increased amounts of immature, band neutrophils are found in the peripheral blood. While this is common in many species, it rarely occurs in adult horses even in severe inflammatory situations.

Age and gender are further factors that might potentially influence blood counts. With increasing age, a relative increase in the percentage of neutrophils has been observed in horses, as the total number of leukocytes decreases [42,45]. Foals also possess higher absolute neutrophil counts than adults do, especially fillies do. This gender variation is lost when the adult age is reached [29].

### 3.2. Granulopoiesis

The pathways regulating hematopoiesis in different species are conserved, as a comparison between teleost fish and mammals has shown [57]. The process of granulopoiesis takes place in the bone marrow in all vertebrates except fishes, where the kidney takes over this function. It is commonly divided into the two steps of lineage determination and committed granulopoiesis. The granulocyte lineage determination describes all events from the hematopoietic stem cell to unipotent cells. Hereby, the stem cell, which could give rise to all different types of hematopoietic cells, transforms into either a common lymphoid progenitor or a common myeloid progenitor. This is regulated by BTB and CNC homology (BACH) and CCAAT/enhancer-binding protein (C/EBP) transcription factors. BACH induces lymphoid or erythroid genes and simultaneously represses C/EBP, whereas the latter one induces myeloid genes and inhibits BACH [58]. The common myeloid progenitor on one hand gives rise to cells becoming thrombocytes, erythrocytes, or mast cells, on the other hand to the uniform myeloblasts. These myeloblasts then mature during the committed granulopoiesis and can form granulocytes and monocytes [35], depending on the balance between C/EBPα and PU.1. A high C/EBPα expression leads to granulopoiesis, PU.1 to monocytes [59]. Another factor, the transcription factor growth factor independent-1, not only represses genes to make progress in stem cell differentiation, but also inhibits monocyte-promoting transcription factors and thereby becomes essential for the differentiation into neutrophils [60]. The first step after the myeloblast is the promyelocyte, in which besides a more prominent cytoplasm also granules begin to exist. At this stage, the division between basophil, neutrophil, and eosinophil granulocytes is already possible. As soon as the nucleus begins to shrink, the stage of myelocytes is reached. In the following metamyelocytes the nucleus becomes kidney shaped. Afterwards, the band granulocytes develop and finally mature neutrophils with segmented nuclei emerge. Distinct factors are involved in the differentiation processes beyond the promyelocyte stage. Two of them are C/EBPε, necessary for the transcription of granule proteins, and PU.1, whose expression increases during maturation. C/EBPβ, C/EBPγ, C/EBPδ, and C/EBPζ values increase from the metamyelocytes on, whereas C/EBPα diminishes from the myeloblasts onwards [61].

The release of neutrophils from the bone marrow is calibrated by CXCR4 and CXCR2. CXCR4 ensures the maintenance of neutrophils within the bone marrow, while binding of CXCR2, G-CSF receptors, or Toll-like receptors trigger their release [62,63]. While human neutrophils completely mature inside the bone marrow, there are indications for a completion of maturation outside the bone marrow in mice [64] and rhesus macaques [65].

## 4. Morphology and Composition of Neutrophils

If compared to other cells, granular leukocytes may, in most mammalian species, easily be recognized due to their multilobed nucleus and then further distinguished into neutrophils, eosinophils, and basophils according to the staining behavior of their granules. The spherical cells possess an irregularly lobulated nucleus, which lead to the term polymorphonuclear cell. Moreover, immature and mature neutrophils are identified by their banded or segmented nucleus, respectively. Obviously, a species-dependent difference in the numbers of banded and segmented neutrophils circulating in the blood exists (Table 1).

The neutrophil’s cytoplasm is filled with granules staining with neither eosin nor hematoxylin. However, some species have cells equivalent to neutrophils, but are stained with eosin and therefore called pseudoeosinophils or heterophils. Such cells can be found in birds, reptiles, amphibians, and some fishes, but also in a few mammals [66]. Examples include guinea pigs, rabbits, and chinchillas [1]. Besides heterophils, amphibians additionally have a small number of cells that are stained as neutrophils [67].

Further organelles inside the cytoplasm of neutrophils and heterophils are mitochondria, a Golgi apparatus, ribosomes, and rough endoplasmic reticulum, each in relatively small amounts. Nevertheless, there are differences in the morphology of neutrophils between species, which go beyond the well-known distinction of neutrophils and heterophils. Examples of the diversity of neutrophil morphology are shown in Figure 2.

### 4.1. Size

Neutrophils are spherical in suspension and estimated at sizes of 7–7.5 µm in different species and 8.85 ± 0.44 µm in humans [68,69]. However, most described sizes of neutrophils lie between 10 and 20 µm in diameter (Figure 3) [35]. An exception are amphibians, where it may reach up to 30–32 µm [66]. This increased size, compared to the size of neutrophils in suspension, is due to a flattening of the cell after adherence, as it occurs for instance in blood smears. Neutrophils tend to in- or decrease more likely in their amount than in size, but some influences lead to variations. Therefore, the mean neutrophil volume (MNV) may be measured. This dimension is elevated in inflammatory diseases, as well as during infections or trauma. The same was observed in humans and pigs with myocardial infarcts, where additionally an investigation of the contributing subtypes of neutrophils was performed. This revealed that the increased MNV is caused by two factors. On one hand, a higher percentage of band neutrophils, which have a larger MNV compared with non-activated mature neutrophils. On the other hand, activated mature neutrophils also have a higher MNV. This dose-dependent increase of MNV was shown by in vitro lipopolysaccharide (LPS) stimulation of pig blood [70]. However, LPS stimulation of equine neutrophils resulted in a shift from medium sized towards both smaller and larger neutrophils [71]. A transient, rapid swelling of neutrophils during activation, induced by different chemotactic agents, was additionally observed in rabbit neutrophils. The authors suggest membrane ruffling and pseudopodia formation as reasons [72]. A few days after stimulation, large neutrophils may appear, presumably due to skipped maturation stages. For instance, larger neutrophils were found in horses two to three days post endotoxin infusion [73]. A similar picture of giant neutrophils after abnormal maturation was seen in AIDS patients [74]. These findings suggest that variations in size may occur due to disease, accelerated maturation, or activation in multiple species.

### 4.2. Nucleus

#### 4.2.1. Core Shape

Regarding the lobulation of the neutrophil nucleus in mammalian species an immense variation from the common round shape of other cells can be found. Further structural components, besides the lobes, are thin connecting segments and sometimes nuclear appendages. These appendages enable a differentiation of sex. Drumstick forms, equivalent to Barr bodies, are more often present in females and tag- or hook-like forms dominate in males [76]. This observation was not only made in humans, but for example also in horses, cattle, buffalos, and dogs [1,76,77,78]. Example pictures for this observation in neutrophils from mantled howler monkeys are presented in Figure 4. In contrast, no certain gender-specific appearance could be confirmed in mice, rats, guinea pigs, and rabbits [77]. Furthermore, a difference between the morphological appearance of mature and immature neutrophils needs to be made. From the myelocyte onwards, an indentation of the round shape occurs. Neutrophils have reached the state of band cells when at least half of the nuclear diameter is indented [79]. Maturation is fulfilled as soon as one thin filament is present. This multilobulation along with the contained condensed chromatin results in an inability of neutrophils to divide [35].

However, intra- and interspecies variations in segmentation exist (Figure 5). Lobe numbers of human neutrophils usually range from two to five, with about 50% three-lobed in health [80]. Nevertheless, this amount can change if cells are activated. A higher segmentation can also be the consequence of drug treatment (e.g., G-CSF or corticosteroids), nutrient deficiencies like folic acid or vitamin B12, associated with certain diseases or artificially after long storage of blood [1,80]. In comparison to humans, such hyper-segmented nuclei are normal for animals like guinea pigs, rabbits, and camels. Neutrophils of mice and rats appear hyper-segmented due to their numerous indentations, but can also present as ring forms [1]. By contrast, many birds and fishes have fewer lobes than human neutrophils. Most other mammals are comparable to humans regarding segmentation [81]. These differences already suggest that the often-used definition of hyper-segmentation as the presence of more than five lobes [82,83] cannot be used for all species. Unlike in other animals, equine neutrophils often have more than five lobes (Figure 2 and Figure 5). The additional long filaments connecting the lobes, the overall long and irregular nuclei, with many of them even folding back on themselves, make accurate counting of lobes in horses difficult [83]. This can lead to problems if, for example, cell nuclei are counted automatically and software-based, if this software has been established for human neutrophils [84].

The above described typical picture of a multilobed neutrophil nucleus, lying at the center of the cytosol, cannot always be maintained when comparing animal species. Especially nonmammalian species differ greatly in their nuclear appearance. While bird and amphibian heterophils contain a multilobed nucleus, the majority of invertebrates along with some reptile and fish species have a nucleus with mononuclear shape [66,85] (Figure 2). The heterophil nucleus in snakes and turtles is round, but some lizards show cells with two lobes instead [86,87] (Figure 2). Equally, the form in fishes ranges from kidney-shaped in sharks to two- or three-lobed (e.g., trout). Meanwhile, the already apparent segmentation in amphibians even increases during hibernation of frogs [66]. These immense variations seen within the animal kingdom were suggested to be due to the different tissue densities existing between species [86].

#### 4.2.2. Factors for Segmentation

One of the benefits of a segmented nucleus is an enhancement of nuclear flexibility and thus cell motility when migrating through tight spaces within tissues [81,88]. It enables an elongation of the nucleus and could explain why neutrophils with hypo-lobulated nuclei were deficient in tests of migration through membranes [89]. This coincides with the fact that, with a three- to fourfold higher velocity than other leukocytes and dendritic cells, neutrophils are the fastest cell type [90]. Another suggested advantage is the freed-up space in the cytoplasm offering more places for phagocytosed pathogens [86].

Many factors allowing this nuclear feature have been discussed. Chromatin is present as condensed heterochromatin over long distances, called super contraction. This goes along with a rearrangement of the nucleus, in which for example nucleoli and the heterochromatin relocate to the nuclear lamina [85]. Additionally, the nuclear envelope exhibits some modifications in neutrophils, which are conserved across species [88]. As such, the components lamin A/C and LINC (linker of nucleoskeleton and cytoskeleton) decrease compared to other cell nuclei, but lamin B becomes the highest expressed lamin [88]. A reduced amount of lamin A/C not only results in more flexibility, but an overexpression in neutrophils rather leads to nuclear rounding [91]. Furthermore, the amount of lamin B receptor (LBR) increases [88] and coincides with higher segmentation in a dose-dependent manner [81,92]. Nevertheless, the LBR seems important for the generation of multilobed nuclei [85] mostly during maturation [86], as low levels were found in mature human granulocytes [93]. This receptor is located in the inner nuclear membrane and demonstrates a conserved protein sequence across vertebrates [81]. If mutations in the receptor occur, hypo-lobulated and ovoid nuclei appear, a condition referred to as Pelger–Huët anomaly [92]. While the morphology of neutrophils is affected by this, phagocytosis, chemotaxis, and respiratory burst remain normal in humans [81]. Conflicting findings in LBR deficient mice neutrophils with regards to chemotaxis and the respiratory burst were observed [89]. Nevertheless, this could either be caused by species-specific differences or by variations of the model [88]. However, this anomaly has not only been reported in humans, but also in horses, dogs, cats, rabbits, and mice [81]. It is therefore possible that other factors could contribute to the regulation of LBR, but also to the segmentation in general. Such proposed factors are histones stabilizing the heterochromatin [93] and low concentrations of vimentin [86]. Further factors influencing the neutrophils’ flexibility found in the cytoskeleton include contractile myosins and microtubules [88]. Finally, the suggested influence of the particular nuclear envelope composition on neutrophil functions indicates possible interspecies variations during processes concomitant with deformation and remodeling of the nucleus, such as migration and NET formation [88]. Therefore, it would be interesting to compare components of neutrophils and heterophils, as well as of multilobed and unlobed granulocytes.

### 4.3. Cytoplasm and Granules

The color of the cytoplasm of neutrophils and heterophils in Diff-Quick staining varies only slightly from colorless to light eosinophil or pale blue. If a large proportion of the cytoplasm is covered with granules, such as in cattle, it may appear eosinophilic (Figure 2 and Figure 5) [1]. Meanwhile, the morphology of granules between species differs greatly. While nonmammalian neutrophils have coarse, irregular formed and sized granules, mammals usually have fine granules. Latter granules remain invisible or stain slightly eosinophil with Diff-Quick dyes. Species with pseudo-eosinophils, in which large eosinophilic granules are inside the cytoplasm, are an exception. Additionally, cattle have such large granules, but in addition to the fine ones [1]. Fish cells are referred to as either neutrophil or heterophil depending on their size of granulation [87]. The controlled release of these granules enables the activation of resting neutrophils to exert their effector cell function [94]. In general, three different types of granules present inside the neutrophils are distinguished, due to their variable content of enzymes and their time point of formation. The primary or azurophil granules already appear in early promyelocytes [95] and are typically rich in myeloperoxidase (MPO). As the production of this enzyme stops in myelocytes, the later on produced granules are peroxidase negative [94]. This type of granule is furthermore divided into specific or secondary and gelatinase or tertiary granules. The latter only form in metamyelocytes and band cells, are characterized by their high gelatinase content [96,97], and are not present in all species. Meanwhile, the secondary granules present with low or lacking gelatinase but high lactoferrin contents [97]. Additionally, secretory vesicles are found inside the cytoplasm of segmented neutrophils. They are formed by endocytosis after the cessation of granule formation in mature neutrophils [95]. In some species, further types of granules were found. As an example, human neutrophils contain ficolin-enriched granules [98], whereas cattle present so-called large granules to store oxygen-independent microbicidal agents [99]. This already points towards substantial differences in protein contents between the granule types, as well as further variation within animal species. As these proteins are essential for the antimicrobial and digestive characteristics of neutrophils [63], heterogeneities may illustrate divergent roles of neutrophils among species. Comparisons between species remain difficult, as some components have not been studied in all species and additionally laboratory techniques vary regarding isolation and processing of the blood cells [1,100]. Nevertheless, an overview about granule components of neutrophils and heterophils in different species is given in Table 2.

Myeloperoxidase (MPO), the characterizing enzyme of primary granules, is present in most animals except birds and some fishes [1,101,102]. In fish, the presence of MPO is used to distinguish between species with neutrophils and those with heterophils [102]. Nevertheless, even within the animal species in which MPO is present, the amounts vary notably. For instance, it is reduced in cattle, mice, and cats [99,101]. The amount in humans (109 ± 44 × 10^−3^ ΔOD/min per 5 × 10^6^ PMN) is in a comparable range to squirrel monkeys and dogs [101]. The antimicrobial peptide lysozyme can be found in all types of granules, but with highest amounts in the secondary ones [103]. Of the species investigated by Rausch and Moore, humans and chicken show the highest amounts within their neutrophils or heterophils, respectively. The activity in other species like dogs and horses only accounts for 10–20% of the one found in humans [101]. Meanwhile, lysozyme is absent in monkeys, cattle, goat, sheep, cat, and hamster neutrophils [1,101]. Another granule component of some animals is alkaline phosphatase. It is not expressed in neutrophils or heterophils of mice, birds, cats, rhesus monkeys, and some fish species [1,101,102]. The amounts in humans are 11 ± 4 × 10^−3^ U per 5 × 10^6^ PMN comparatively low, as values for instance in horses reach 471 ± 76 × 10^−3^ U per 5 × 10^6^ PMN [101]. Furthermore, the pH of maximal enzyme activity differs between species, as well as the requirement of heavy metals. As an example, Mg^2+^ is required in humans, while Zn^2+^ is necessary in cattle and horses [101]. The lactoferrin, primarily stored in secondary granules, shows considerable variations among mammals. Even though it was detected in all investigated species, the amounts in carnivores and primates were higher than in herbivores [104]. The consequent staining was the least in cows and in size the largest in rabbits. Interestingly, milk lactoferrin concentrations correspond to the amounts in neutrophils of the respective species [104]. Located in the matrix of azurophil granules of all investigated species is β-glucuronidase. It was detected in variable amounts, with the highest values in human neutrophils and very low ones for example in pigs [94,101,105]. The species-specific differences in the proteins contained in the granules should therefore be considered in studies of neutrophils. This is especially important when they are used as markers for neutrophils or NETs.

The major classes of antimicrobial peptides in mammals are defensins and cathelicidins. Principally three categories of defensins exist in vertebrates. α- and β-defensins are the more common ones [118], while θ-defensins are only expressed in granulocytes of some monkeys, for instance rhesus macaques and orangutans [121]. The α-defensins are the ones typically found in mammalian neutrophils, but they do not exist in all species. For instance mice [119], horses [117], pigs [118,122], cattle [119], birds, and reptiles [116] lack α-defensins in their granules. Reptiles and birds furthermore do not produce α-defensins at all, even outside of neutrophils [116]. Meanwhile, β-defensins are primarily produced in epithelial cells [119] but are nevertheless present in neutrophils of some species. Examples are birds and some reptiles, but also cows [116,120]. Pig neutrophils express a β-defensin gene, but no defensin peptide has been documented in their granulocytes [122]. Equine neutrophils do not contain defensins at all but possess equine neutrophil antimicrobial peptides (eNAP-1 and -2) instead, which seem closer related to cytokines and microbial protease inhibitors than to actual antimicrobials [117]. At least one variant of the other big class of antimicrobial peptides (AMPs), namely cathelicidins, exists in every mammalian species investigated so far [148]. A single one was found in humans, dogs, cats, mice, and rats [147,148]. Meanwhile, multiple exist for example in horses, cattle, pigs, and sheep, with pigs possessing the largest repertoire [117,122,148]. Cathelicidin-like peptides or sequences are also reported in nonmammalian vertebrates like chicken, rainbow trout, snakes, and lizards [116,148]. These AMPs are stored as inactive pre-propeptides in secondary granules and activated upon release through cleavage mediated by elastase or other proteinases [148]. The structure of the functional peptides is highly divergent within and between species, as the sets of genes are different [117,150]. Nevertheless, high similarities were observed between canine, murine, human, and feline sequences [147]. Furthermore, the effects achieved by different types of cathelicidins varies [151], as well as their relative amount in the neutrophils [148].

The diversity of granule contents suggests that the contribution of single antimicrobial peptides and enzymes to the overall microbicidal effect varies between species [122]. A separation of these components into diverse types of granules was necessary to prevent unintended activation or inactivation of interacting substances [152]. As the composition differs among animals, it is evident that the granule types also differ. Interestingly, despite the multitude of granule variations between species, no correlation between certain enzyme contents and the absolute number of neutrophils in the peripheral blood could be detected [101].

## 5. Function of Neutrophils

Starting already in slime molds, neutrophil-like cells are dominant in the fight against pathogens throughout organisms [153]. In animals lacking an adaptive immune system, like insects, survival depends on the functionality of the innate immune cells [154]. However, with quite a range of similarities and differences in development, morphology, and components, a variation also occurring in neutrophil function is obvious. Nevertheless, Canfield et al. claimed that granulocytes possess similar functions in most species. This statement emphasized a common basis of activity: the microbicidal and phagocytic aspects of neutrophil actions [87]. Other conserved antimicrobial reactions comprise degranulation and NET-formation [9].

Resting neutrophils are predominantly not in the blood stream, but rather sequestered in small blood vessels of liver, spleen, lungs, and bone marrow [35]. A chemokine-induced recruitment is sustained among different vertebrates [32,155]. This activation happens due to infections, but also in non-infectious inflammation like in injured tissue after ischemia or trauma [156]. Mature neutrophils may be mobilized from the marginal pool, but in many cases, also immature forms are released from the bone marrow. This reaction to inflammatory mediators is detectable in all classes of vertebrates [1,157,158] except amphibians, where band forms are not typical [159]. Another exception is present in cattle, as their production period of neutrophils does not decrease under enhanced request. Combined with their limited capacity of production, neutropenia is common at the beginning of inflammatory diseases in cattle [1], in contrast to the left shift along with neutrophilia in other species.

Once arrived at its destination, the neutrophil needs to cross the vascular endothelium to reach the tissue, without being activated or damaged by this process [5]. This is initialized by changed hemodynamics, leading to margination at sites of inflammation [160]. The stimulated endothelial cells release activating substances and mobilize adhesion receptors such as selectins, so neutrophils can interact with them. This is achieved by rolling of the neutrophils along the endothelium, adhesion, and finally transmigration. During this process, secretory vesicles are relocated to the neutrophil surface, enriching the membrane with β-integrins. As this implements firm adhesion, transmigration is enabled [94]. During this process, neutrophils crawl through the endothelium and basal membrane of blood vessels, either para- or transcellularly [63]. Afterwards, the immune cells follow a chemotactic gradient through the tissue towards the inflamed spot. Such chemo attractants are comprised of complement factors, chemokines, or bacterial components, such as N-formyl- methionine-leucyl-phenylalanine (fMLP) [161]. A particularity of bovines is that besides the inducible chemokines, one (CXCL3) is always expressed in the mammary epithelium, resulting in constant low neutrophilic invasion into the milk, also in health [162]. In general, all these processes ceasing in tissue invasion are developed in neutrophils and heterophils [158].

After transmigrating into the inflammatory area, the neutrophil tries to combat the pathogen. This may be conducted by phagocytosis, a process conserved throughout evolution [153]. The rates by which certain pathogens are ingested may vary between species [163], but the principle remains the same. The neutrophil engulfs a pathogen into a phagosome, which then fuses with vesicles from the endoplasmic reticulum and Golgi complex, as well as with lysosomes to build the phagolysosome [161,164]. Microbes are then intended to be killed in oxygen-dependent and -independent ways. The microbicidal components split the pathogen enzymatically, but also reactive oxygen species (ROS) are involved [153]. Phagocytosis is mediated by receptors, as pathogens are recognized through either opsonins, antibodies, or pathogen-associated molecular patterns [161,165].

Another possible mechanism is degranulation. The varying components mentioned earlier (Table 2) already imply differences in the effects of granule release between species. Hence, the principles of this are summarized with the human example. The release of granules needs tight control to prevent cytotoxic effects of the components where not desired [161]. First, secretory vesicles are released to enable adhesion. The following tissue degradation necessary for migration is enabled by the release of peroxidase-negative secondary and tertiary granules [153]. The last granules to be mobilized are the primary ones containing for example MPO and defensins [5]. Upon pathogen confrontation, the neutrophil can release granules or initiate their fusion with the phagosome [153]. The granule components act via different ways, e.g., by pore formation, interference with metabolic pathways, participation in ROS production, or disruption of biofilms [94,153].

The most recently discovered defense process is NET-formation [2], whereby neutrophils release decondensed chromatin combined with histones and granule components to the extracellular space [2]. Thereby, the web-like formations catch and combat pathogens, but also detrimental effects such as a contribution to certain autoimmune diseases may occur [166,167]. The conservation of this mechanism among species has recently been highlighted in a review by Neumann et al. [9]. Its occurrence is reported in various vertebrate species: mammals, including marsupials, birds, and fish. Additionally, invertebrates can extrude extracellular traps from their phagocytes, and even plants were observed with extracellular DNA traps. Still, there are no reports about extracellular traps in amphibians and reptiles, only hints about heterophil projections in a turtle [168]. Apart from this common ground of building extracellular traps, originating from neutrophils or other cells, similarities have also been described in the composition of these traps, as well as in the provoking stimuli [9]. Nevertheless, if little or no granule protein is present in a species, the NETs structures may differ with respect to the embedded proteins.

Reactive oxygen species (ROS) play an important part in all different ways to combat pathogens. These short-lived radicals and reactive molecules derived from molecular oxygen are produced by different mechanisms. The most important one in neutrophils is initiated by the assembly of cytosolic and membrane-bound components of the NADPH oxidase (NOX2), mediated by bactericidal or immune stimuli, as well as in response to phagocytosis [169]. This assembly enables the directional electron transfer towards oxygen, and thus its reduction to superoxide, a precursor of hydrogen peroxide and further ROS types [170]. This happens either towards the extracellular compartment or towards the inside of phagosomes. Thereby, it explains the ROS involvement in phagocytosis. Additionally, NET-formation is described to be at least partially NOX2-dependent in response to most stimuli, and can thus be circumvented by inhibition of the NADPH oxidase [171,172]. Hydrogen peroxide in turn reacts with other substrates, catalyzed by MPO, into more potent antimicrobials [5]. The production of oxygen radicals by avian heterophils is low compared with mammalian neutrophils [1,163] and may be explained by the missing MPO in avian granules. Chicken heterophils are furthermore unable to produce high amounts of hydrogen peroxide or superoxide anion [173,174] and seem to mainly act by non-oxidative mechanisms [158]. As MPO deficiency in humans is rarely associated with serious infections despite the poor killing of many microorganisms, a backup mechanism in NOX2 is assumed [170,175]. The process of ROS production and release is critical for the battle against pathogens [153] but needs a tight regulation. Rampant release results in oxidative stress and tissue damage, for instance in acute lung injury or in organ failure during sepsis [176].

Apart from these microbicidal functions of neutrophils, also immunomodulatory effects exist. As such, they release components and factors to thereby contribute to the communication between different types of immune cells [177]. For example, the release of cytokines influences macrophages to differentiate into pro- or anti-inflammatory cells [178] and some granules stimulate the chemotaxis of T-cells [179]. The neutrophils’ production of a range of chemokines explains their chemotactic and immunomodulatory effect towards other leukocytes, and furthermore their modification of angiogenesis or influence on tumor growth [180,181,182]. The repertoire of chemokines and their receptors is highly conserved between humans and mice [155]. However, others claim significant differences in the genomes of these two species regarding cytokines [183]. However, even chickens possess orthologues to most human chemokine receptors and produce chemokines of all known human classes [155]. Hereby, it should be kept in mind that the complete genome has been evaluated for chemokine and associate receptor genes in this study. Hence, not all of these may originate by neutrophils, because all cell types are able to produce chemokines to a certain extent [184]. Additionally, frogs and fish produce these signaling proteins, the latter with noteworthy variation between species [155]. A lineage-specific generation within vertebrate species is obvious, however, the receptors seem more conserved than the chemokine genes [183,185,186]. Meanwhile, invertebrates lack this modulation system, suggesting an origin at the evolutionary emergence of vertebrates [155,185]. Via signaling peptides, vertebrate neutrophils also modulate their own behavior. They for instance induce further chemotaxis, expression of chemokine receptors, or suppress apoptosis [153].

In regard to apoptosis, neutrophils are generally considered to have a short lifespan with proposed circulating half-lives of 4–19 h [187,188]. One study claimed a more extended life of 5.4 days in healthy people, corresponding to a half-life of 3.8 days. In addition, a species difference to mice was found, since their neutrophils revealed a lifespan of only 0.75 days [189]. However, doubts regarding the model itself, as well as concerning the results emerged. These doubts arose because bone-marrow neutrophils were probably included in the labelling, leading to overestimated lifetimes [187,190]. Nevertheless, the neutrophil’s lifespan may be extended after activation to ensure constant presence during inflammation, or due to pathogens modifying neutrophil mechanisms [191,192]. At the end of their being, neutrophils undergo apoptosis once the antimicrobial tasks are fulfilled. Apoptotic neutrophils are locally incorporated by macrophages or dendritic cells, whereby macrophages are induced to release anti-inflammatory signals [193]. Another way out of the tissue is reverse transmigration back into the circulation, which has been detected in zebrafish first [194], but meanwhile also in mammals. Dying granulocytes are furthermore cleared in the bone marrow, liver, and spleen. An additional exit way from the circulating blood is towards the environment via urine, oral mucosa, the ocular surface, or pus [161]. Purulent lesions in mammals differ enormously from the ones in birds and reptiles, even macroscopically. In mammals, liquid exudate or abscesses develop. By contrast, birds and reptiles produce caseous pus including the necrotic heterophils, enclosed in granulomas [195]. Common to all species is the importance of these strategies to clear neutrophils, as granule components are potentially damaging, and abundant neutrophil action can lead to tissue damage and exposure of autoantigens [156].

In summary, neutrophils are key players of the immune defense, with many similarities in their execution routes between species (Figure 6). In nonmammals, the heterophil performs an essential proportion of bacterial defense, just like the mammalian neutrophil, but is contrarily also strongly supported by monocytes. Consequently, bacterial infections in nonmammalian species may go along not only with heterophily, but also a huge increase in circulating monocytes [87]. Thereby, the heterophilic response in avian species and reptiles are more alike to each other than to the neutrophil response of mammals [195]. Different granule compositions may explain that, despite both a reduced ROS production and phagocytosis, chicken heterophils still manage to exhibit a killing ability comparable with human and dog neutrophils [163]. Another explanation of differing responses is the varying equipment of neutrophils from different species with receptors. This has recently been reviewed by van Rees et al. [196] comparing mice and humans. Differences arise even in as closely related beings as nonhuman primates and humans, moreover between the distinct primate species [65]. A familiar example for species specificity is the fMLP receptor, present with one or two subtypes in some animals. Such receptors requiring variable intensities of stimulation were found in humans, rabbits, guinea pigs, mice, and nonhuman primates. Horses also respond to fMLP, but unlike other species only with secretion and not with chemotaxis. Meanwhile, no reaction occurs in bovine, ovine, porcine, galline, canine, and feline neutrophils [100,163]. Investigating such neutrophil responses in different species is fundamental for deciding if results can be extrapolated to other species or not. Even though for example bovine neutrophils are similar to the ones derived from rodents or humans, some differences with an impact on certain functions exist [152].

## 6. Diseases of Neutrophils

Neutrophils play an important role as first line of response in infectious diseases. However, beyond their part in immune defense, neutrophils can also show functional disorders. Besides influencing factors like stress level and age, neutrophils can get infected themselves or exhibit malfunction due to gene mutations [1]. In case of anaplasmosis disease, neutrophils get infected and host the pathogen. Thereby, the Gram-negative bacterium *Anaplasma phagocytophilum* [197] is transmitted to animals or humans by the common tick *Ixodes* spp., using wildlife animals as intermediate host [198]. Mainly neutrophils get infected by this bacterium that only multiplies in host cells [197,198]. Thereby, *Anaplasma* not only escape from the immune system by hiding, but also modify neutrophil functions towards restricted movement and lowered phagocytosis rate [199]. Further modifications include hindrance of superoxide production, reduced adherence and transmigration, and retarded apoptosis [192,199]. Another pathogen with the ability to alter neutrophil functions is *Chlamydia trachomatis*. This microbe mostly leads to subclinical persistent infections in humans, as it evades the immune response by paralyzing neutrophils at the site of infection. Therefore, the microbe enables its transmission from one host cell to another without being killed by neutrophils. This paralyzation includes failure of NET-formation, oxidative burst, chemical-mediated activation, and a reduced cell death [200]. Furthermore, infections of neutrophils can be temporary when neutrophils are used as Trojan horse. Examples for this would be infections with *Leishmania major*, *Chlamydia pneumoniae* or *Brucella abortus*. These intracellular microbes use the uptake into neutrophils to induce apoptosis. Since apoptotic cells are phagocytosed by macrophages without inflammatory reaction, the goal of the bacteria to reach these cells without activating the immune system is achieved [201,202,203].

Apart from pathogen-mediated dysfunctions, numerous gene mutations have been observed in neutrophils, which may be classified into four classes of neutrophil disorders: quantity, granules, chemotaxis, and killing [32]. A selection of disorders which not only affect humans, but also have been reported in animals is presented in Table 3. The effects differ immensely between affected genes, but sometimes also between species within the same mutation. An example with conserved effects across species is the leucocyte adhesion deficiency. This autosomal recessive mutation in the CD18 allele leads to decreased phagocytosis and degranulation, as well as failure of transmigration [204,205]. Consequently, the affected dogs, cows, and humans typically have a severe neutrophilia, but no obvious tissue infiltration with neutrophils. This results in recurrent infections without apparent pus formation [32,205].

The influence of neutrophils on cancer diseases is of increasing interest. Generally, five different types of neutrophils are described in the state of cancer, of which three are found in the peripheral blood [206,207,208]. Among them are high-density neutrophils (HDNs) with a mature phenotype, acting cytotoxic and against tumor-progression. The heterogeneous low-density neutrophils (LDNs) are pro-tumor and immunosuppressive. They can be further subdivided into mature LDNs and immature granulocytic myeloid-derived suppressor cells (G-MDSCs). Nevertheless, these types are plastic as at least HDNs change into LDNs if influenced by transforming growth factor-β (TGF-β) [207]. Additionally, the HDNs of multiple myeloma patients, including people that are still in the preceding asymptomatic phase, are impaired concerning phagocytic activity and oxidative burst. As the resulting immune-suppression by HNDs is not present in according murine models, a difference in cancer-related myelopoiesis between these species was suggested. Thus, also HDNs are able to support tumor progression and susceptibility to infections at least in human multiple myeloma [209]. In addition, the group of G-MDSCs is composed of various subsets in the tumor microenvironment of multiple myeloma patients: eosinophils, basophils, and three neutrophil maturation stages. Only recently, the expression of the Lectin-type oxidized LDL receptor 1 (LOX-1) was detected as a specific marker to distinguish the PMN-MDSCs from other neutrophil types in humans, but not in mice [210]. Interestingly, a progressive immunosuppressive potential ensuing from immature to mature stages of G-MDSCs was detected. The pro-tumorigenic response goes along with induced transcription of inflammatory signals and an inhibiting effect on T-cells. This is due to a molecular reprogramming of the mature neutrophils, probably influenced by TGF-β [211]. Another mechanism involved in tumor progression is autophagy, leading to cell survival under stress conditions if upregulated in neutrophils [206,212]. In multiple myeloma patients, factors released by neoplastic cells trigger autophagy in HDNs. This modification in response to environmental changes could explain why HDNs in these patients are pro-inflammatory and survive longer [213].

Such a stimulation by substances derived from cells of the tumor microenvironment leads to anti-tumor (N1) and pro-tumor (N2) subtypes in the now tumor associated neutrophils (TANs) [214]. This is of special importance, as neutrophils are the most abundant inflammatory cells in solid tumors and almost every tumor produces chemokines to regulate the TANs in its environment [206]. Nevertheless, it has to be kept in mind that N1 and N2 TANs have only been observed in mice so far and that their presence in other species is uncertain [206]. The N1 TANs act against cancer by activating further immune cells and by releasing cytotoxic ROS or proteases. Meanwhile, the long-living N2 TANs are not cytotoxic, but display pro-angiogenic, -metastatic, and immunosuppressive effects [206,208,214]. Neutrophils may also induce tumors, as through DNA damage by ROS [215]. Furthermore, several studies describe a correlation between NETs and cancer. For example they promote metastasis [216,217] or cause cancer-associated thrombosis [218,219].

It is suggested that different immune cells together eradicate small colonies of tumor cells at the beginning of disease [220]. However, in a more advanced state, neutrophils contribute to immunosuppression for example by their influence on T-cells. Thus, for instance CD8 T-cells are induced to undergo apoptosis and limited in their proliferation [221]. The mature population of G-MDSCs also decreases the cytotoxicity of T-cells [211]. On the other hand, TANs can also act tumor-suppressive by influencing T-cells. They induce macrophages to produce more IL-12, which in turn causes unconventional T-cells to release IFNγ. This pathway leads to resistance against murine and some human sarcomas [222].

Overall, cancer diseases underline the importance that neutrophils exhibit also in diseases other than infections. They are not only crucial for understanding the pathogenesis and to develop therapies targeting deranged neutrophils, but also helpful for prognosis.

## 7. Influence of Temperature

Several effects of changed body temperatures on neutrophils have been observed in different species. For instance, hypothermia leads to decreased neutrophil counts in the peripheral blood of mice through deferred differentiation and maturation [228]. Additionally, neutropenia induced by irradiation could be reversed faster under increased body temperatures (39.5 °C vs. ~37 °C in mice) [229]. Due to these influences of core temperature in mice, one could assume a connection between physiologic ranges of body temperature and neutrophil counts in different species, but we could not find any correlation. Meanwhile, a halt of apoptosis, observed in human neutrophils cultured at 15 °C [230], suggests more extensive impacts of temperature beyond an influence on cell numbers. In contrast to low temperature, fever accelerates the apoptosis rate at a neutral environment, but at the same time increases neutrophil survival if low pH is present. This is opposed to the reaction of lymphocytes to fever, as these cells die in low pH settings [231]. Fever additionally seems to maintain neutrophils inactivated as long as they are circulating. However, it does not influence the ones that have migrated into inflamed tissues, as fibronectin-interacting granulocytes react differently than suspended ones under higher temperatures [232]. These effects were seen in TNF-alpha induced signaling, adhesion and delayed apoptosis, as well as gene transcription. Meanwhile, the release of NO and reactive oxygen species is significantly enhanced in adherent human neutrophils during fever-like temperatures [233]. Further influences of temperature on functions were suspected regarding NET-release. Different ambient temperatures influenced the release of NET-like structures by HL-60 cells, a human leukemia cell line, to differing extents. Thereby, cold shock with subsequent rewarming to body temperature resulted in most obvious alterations of DNA similar to NETs [234]. Additionally, physiologic body temperature and hyperthermia like during fever (40 °C) accelerated the initial phase of NET-release compared to human neutrophils kept under hypothermic conditions. This effect, which was only observed until the start of chromatin expansion, is suggested to result from greater enzymatic activities at higher temperatures [235].

Another interesting impact of temperature on neutrophils can be suspected in hibernating and ectotherm animals. During both deep and shallow torpor of hamsters, leukopenia and neutropenia in circulation occur, which goes back to normal upon arousal. This impact of climate is reproducible in rats and hamsters through forced hypothermia [236]. A well-investigated ectotherm species is fish. However, fishes not only have variations in white blood cell counts with changing water temperatures, but also demonstrate opposed changes in different fish orders. For example, Atlantic salmon possesses higher percentages of neutrophils in peripheral blood leukocytes at 18 °C compared to lower temperatures. By contrast, their proportion of neutrophils is not temperature-dependent in head-kidney leukocytes [237]. Meanwhile, circulating granulocyte numbers double in carps at hypothermic conditions, whereas the amounts in the kidney decrease [238]. No effect on blood leukocyte numbers was observed in Atlantic halibut [239].

Since body temperature is variable in different species and it cannot be excluded that it affects the function of neutrophils, this should be taken into account for in vitro experiments.

## 8. Cell Culture and Storage

Due to their short life span, fresh blood-derived neutrophils are limited for longer investigations and it is recommended to finish experiments within 8 h after blood collection. Hence, human and mouse cell lines have been established, with cells that mimic neutrophils as close as possible and that display a neutrophil-like differentiation in cell culture.

Human neutrophil cell lines include HL-60 cells, NB4 cells and the PLB-985 sub-line (Table 4). HL-60 is a myeloblastic tumor cell line [240] resembling normal promyelocytes [241] which can be, besides monocytes, macrophages, and eosinophils, induced to differentiate towards segmented neutrophils. This differentiation may be achieved by various treatments, for instance with dimethyl sulfoxide (DMSO), all-trans retinoic acid, or hypoxanthine (reviewed by [242]). Thereby, DMSO was reported to give the best combination of viability, function, and expression of neutrophil markers [243]. These differentiated cells resemble neutrophils, but significant deviations from normal promyelocytes and between passages appear. As higher passages have less prominent azurophilic granules while large cytoplasmatic vacuoles emerge, and thus resemble early promyelocytes, a selection for immature cells or a dedifferentiation throughout passaging was suggested. Furthermore, decreased staining of ER and Golgi imply a decreased protein synthesis of this cell line [241]. The further deficient expression of late neutrophil specific genes, along with defective chemotactic responses towards several stimuli, argue against the HL-60 cell line [244,245]. Another counterargument is the limited usability for examinations of host–pathogen interaction. The differentiated HL-60 cells exhibit less antimicrobial activity, ROS production, and NET formation compared to human blood-derived neutrophils when challenged with *Staphylococcus aureus* [246]. Nevertheless, these cells share many characteristics with mature neutrophils [244]. The promyelocytic and leukemic cell line NB4, which may also be induced towards macrophages, resembles mature neutrophils but shares the defects in gene expression and chemotaxis seen in HL-60 [244,245]. Another choice could be PLB-985, a sub-line of HL-60 with slightly different characteristics, but that also does not show all neutrophil properties [243].

In mice, the spectrum of cell lines is a bit broader. A multipotent progenitor cell line called EML (erythroid, myeloid, lymphoid) is developmentally blocked, but may be induced towards early promyelocytic cells (EPRO) through a combination of retinoic acid, interleukin-3, and GM-CSF. These cells can afterwards be further differentiated to mature neutrophils. Related are the murine promyelocytic cells (MPRO), which mature by the influence of retinoic acid and were also found with neutrophil-specific gene expressions. Both, EPRO and MPRO cells, are precise models of mature murine neutrophils, as they exert full functional responses like chemotaxis, phagocytosis, and respiratory burst [245]. Similar expression profiles as in MPRO cells were detected in SCF ER-Hoxb8 mouse myeloblasts, pro-neutrophils that are reactivated by removal of β-estradiol from their growth medium. The 32Dcl3 cells are further myeloblasts which mature by addition of G-CSF, even though it simultaneously induces apoptosis in many of them. As all these above-mentioned murine cell lines express complete maturation markers of neutrophils, but are dependent on different factors for maturation, they all represent different models of myelopoiesis [244]. Another widely used model is the amoeba *Dictyostelium discoideum*, as it crawls in a similar manner as neutrophils and is alike regarding chemotaxis [254]. This makes it a good model for investigations of neutrophil motility, as performed for example during a race against HL-60 cells with worldwide contributions [254].

As all these cell lines and models do not manage to fully reflect neutrophil properties, alternative attempts to preserve neutrophils by freezing have been made. Numerous publications about limited success with human granulocytes can be found. Neutrophil numbers decrease about 50–75% and approximately half of the surviving cells do not retain their original morphology and function after thawing [255]. As an example, the phagocytic activity decreases to 27.6% of the original one [256]. Furthermore, the viability of the remaining cells rapidly declines at 37 °C, probably because granules rupture and the released lysosomes act destructively towards the cells [256,257]. Nevertheless, for instance Graham-Pole et al. [255] could store a proportion of the neutrophils with a well remaining bactericidal capacity in liquid nitrogen for about 14 months. Equally, Svedentsov et al. reported a viability of 81% for up to 12 days in undercooling temperature (−10 °C), and in three-quarters of those the ability to phagocytose [258]. Multiple approaches have been tested to improve the outcomes. The blood collection with heparin seemed superior to acid citrate dextrose [256]. DMSO in a concentration of 10% as well as glycerol were used along with other cryoprotectants, such as different pectins [259]. As phagocytosis is inhibited by the presence of DMSO, this needs to be washed away with homologous freeze-dried plasma rather than other media according to Graham-Pole et al. [255]. Furthermore, handling circumstances such as careful centrifugation and working temperatures of 4 °C were beneficial [255]. An interrupted cooling strategy including an adaptation period at −20 °C was more effective regarding phagocytic capabilities compared with a continuous cooling procedure [255]. As an alternative, cryohypobiosis was tested, during which the cells are cooled down to only −10 °C for a reduced metabolism but avoiding intracellular freezing of water [258]. By contrast, equine neutrophils were successfully cryopreserved using equine plasma with 5% DMSO as cryoprotective agent. With a cooling velocity of 1 °C/min between 4 and −70 °C, a loss of only about 20% of leukocytes and 92% viability after thawing were achieved. In addition, the phagocytic function remained unaffected and the ability to generate ROS even increased, suggesting a functional capability for about 6 h after thawing [260]. In a rabbit model, the in vivo functions of neutrophils stored in whole blood at either room temperature or 4 °C for up to three days were compared. Hereby, cell storage at 4 °C for 72 h was possible. The tagged and then reinjected neutrophils were able to circulate and migrate into inflamed tissue, although a recovery period of 1–2 h was necessary after injection. Meanwhile, the samples stored at room temperature were irreversibly damaged after two days [261]. Due to these conflicting results about the feasibility of neutrophil storage, an alternative was suggested by cryopreservation of PMN cytoplasts. Those enucleated neutrophils without granules can be used for research of membrane structures and processes. They otherwise have differing responses compared to complete neutrophils, but show reactive oxygen species (ROS) production, phagocytosis, and limited killing of bacteria. Cytoplasts can successfully be kept at −70 °C for up to five days without any effects compared with fresh ones. These results are considered to be due to the lack of damage by lysosomal enzymes after thawing and to the reduced cell clumping normally caused by nuclear DNA [262]. After all, these results suggest inconvenient outcomes of storage of neutrophils for scientific experiments or even clinical use for transfusions. Some functional aspects may be preservable, but the full picture of a neutrophil’s abilities can so far only be seen in fresh blood-derived cells.

## 9. Conclusions and Perspective

A huge amount of similarities between neutrophil and heterophil granulocytes of different species has been observed, especially regarding principle functions like phagocytosis or the production of reactive oxygen species. Nevertheless, differences in aspects like morphology, components or receptor equipment make it obvious that neutrophils, derived from one species, cannot be expected to fully represent the ones from another. Even minor differences may lead to huge variations, such as a missing receptor type preventing the cascade of reactions occurring in another species. Furthermore, differences in isolation, treatment, and laboratory techniques may limit or even make it impossible to compare results obtained in different studies. A broad comparison of all species is difficult, as different aspects of the whole spectrum of knowledge about the neutrophil are available for every species. Understanding and discovering the variations between animals is valuable for the interpretation of their immune functions. This could be of particular interest to better understand the interaction of different species as hosts and the same pathogen during the host–pathogen interaction. Whether extrapolation of the results derived from one species to another is feasible and may represent the real condition needs to be decided with care and considering the numerous differences that have already been detected.

Evolution has obviously left a fingerprint in the morphology and special functions of neutrophils and heterophils of different species. Nevertheless, there are conserved mechanisms that are similar across species to control infections.

Future research could, however, focus even stronger on analyzing the different responses of neutrophils of different species and thus, for example, understanding host–pathogen interaction in zoonotic diseases.

## Figures and Tables

**Figure 1 ijms-21-04523-f001:**
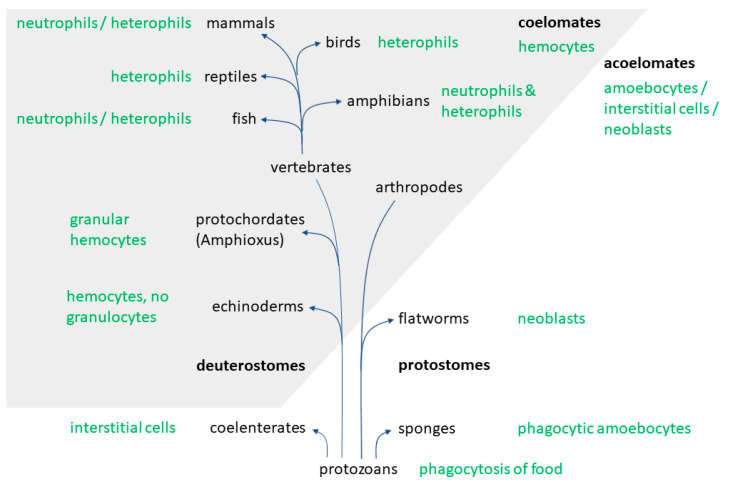
Evolution of neutrophils/heterophils.

**Figure 2 ijms-21-04523-f002:**
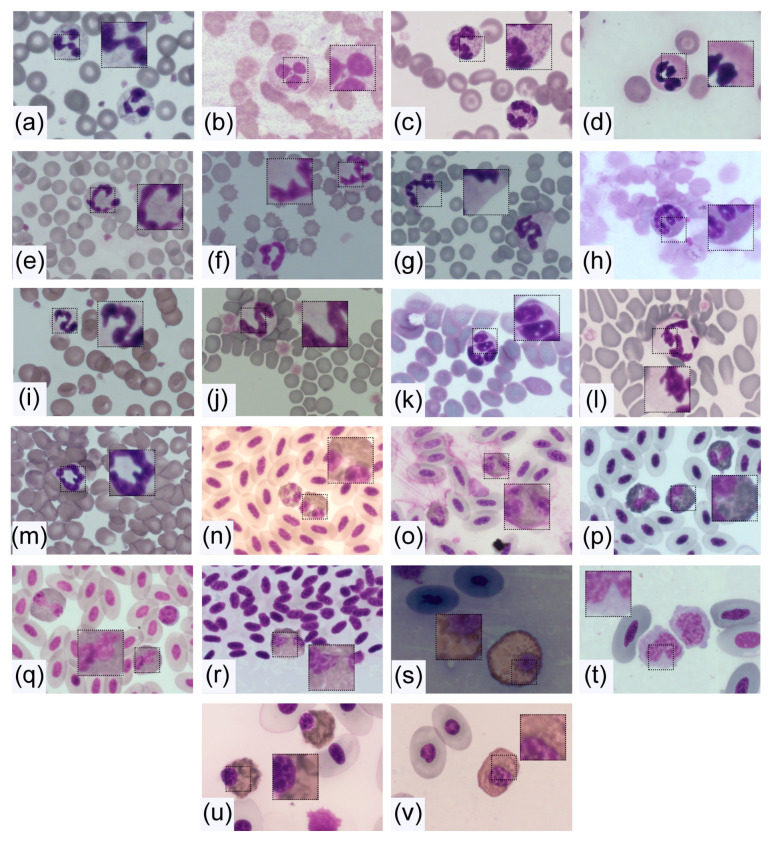
Morphology of neutrophils and heterophils of distinct species. Blood smears were captured after HAEMA fast stain with a Zeiss Axio Imager M2 microscope and all were taken with a 100× objective (Appendix A). The black square (0.6 × 0.6 cm) was cut out and size was doubled to enlarge the cytoplasm and parts of the nucleus. (**a**) Human, (**b**) Geoffroy’s spider monkey, (**c**) Mantled howler monkey, (**d**) Sloth, (**e**) Cow, (**f**) Pig, (**g**) Horse, (**h**) Skunk, (**i**) Dog, (**j**) Cat, (**k**) Squirrel, (**l**) Common opossum, (**m**) Mouse, (**n**) Pauraque, (**o**) Woodpecker, (**p**) Ara, (**q**) Guatemalan screech-owl, (**r**) Common snipe, (**s**) Boa, (**t**) Iguana, (**u**) Common snapping turtle, (**v**) Painted wood turtle.

**Figure 3 ijms-21-04523-f003:**
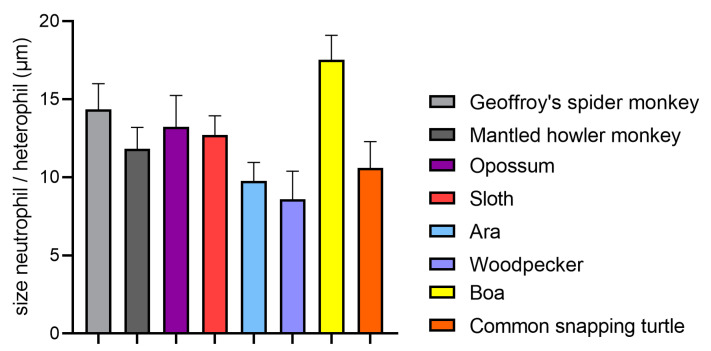
Neutrophil and heterophil sizes in different wild animals. The sizes of 10 neutrophils or heterophils in blood smears (Appendix A) of one blood donor of each species were measured, to get an impression of how the size varies between species. The sizes are given in mean ± standard deviation. In comparison, literature values of domestic animals range from 11 µm in rats, 10–13 µm in birds, and 12–15 µm in cattle [1]. Human neutrophils are described with a size of 7–12 µm [46,75].

**Figure 4 ijms-21-04523-f004:**
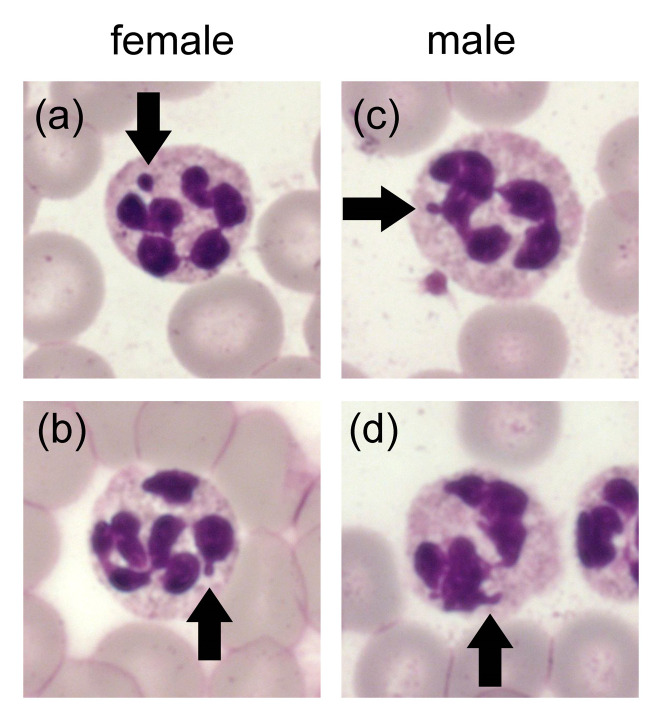
Morphology of neutrophils from male and female mantled howler monkeys is presented. Blood smears were captured after HAEMA fast stain with a Zeiss Axio Imager M2 microscope (Appendix A). Arrows show in (**a**) drumstick form, (**b**) sessile nodules form, (**c**) tag-like form, and (**d**) hock-like form.

**Figure 5 ijms-21-04523-f005:**
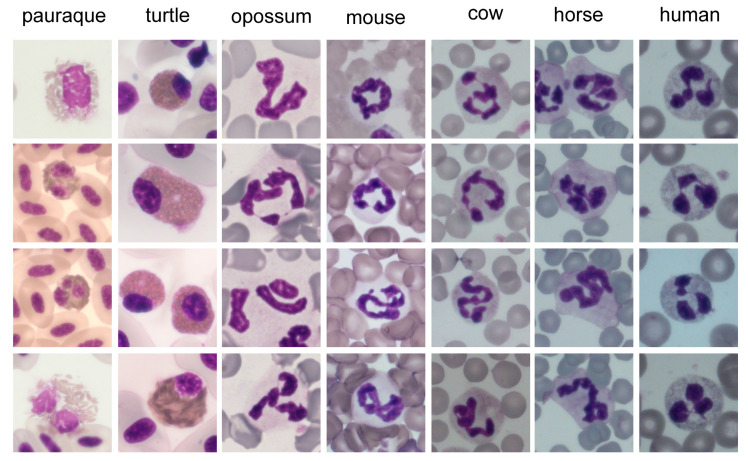
Diversity of nuclei from neutrophils and heterophils of distinct species is presented with four example pictures. Blood smears were captured after HAEMA fast stain with a Zeiss Axio Imager M2 microscope and all were taken with a 100× objective (Appendix A). Then 4 cm squares were cut out and the size doubled.

**Figure 6 ijms-21-04523-f006:**
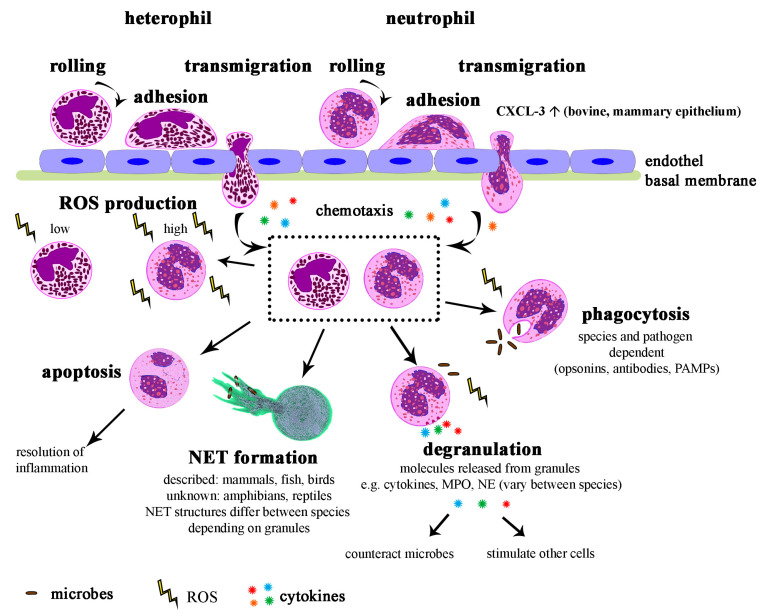
Overview of neutrophil and heterophil functions to counteract infections. After infection of tissue, neutrophils and heterophils are attracted by chemokines to cross cell layers and afterwards to counteract against microbes. Different mechanisms after migration are found in neutrophils and heterophils. If no notable differences are described after transmigration (square), only the mechanism for the neutrophil is shown, which looks the same for the heterophil. MPO = myeloperoxidase, NE = neutrophil elastase, PAMPs = pathogen-associated molecular patterns, ROS = reactive oxygen species.

**Table 1 ijms-21-04523-t001:** Leukocyte and neutrophil/heterophil values in different species.

Species	Leukocytes ^1^	Neutrophils/Heterophils ^1^	Neutrophils/Heterophils ^2^	Band Neutrophils/Heterophils ^1^	Segmented Neutrophils/Heterophils ^1^	References
Human	3–11	1.71–8.25	57–75	0.5–0.77	1.35–8.1	[38,46]
Mouse	2–12	0.4–3.6	20–30	−	−	[1,38]
Rat	2–25	2.4–9.5	12–38	−	−	[1,38]
Horse	3.5–12.1	1.58–8.47	45–70	<0.6	1.6–8.5	[38,41,47]
Cattle	4–13.3	0.6–6.65	15–50	0–0.2	0.6–6	[1,38,47]
Pig	10–22	1–10.34	10–47	0–0.88	2–15	[1,38,47]
Dog	5–17	2.75–14.45	55–85	0–0.45	2.9–12	[1,38,47]
Cat	5.5–19.5	2.48–15.21	45–78	0–0.3	2.5–12.5	[1,38,47]
Chicken	20–30	5–15	25–50	−	−	[37,38]
Budgerigar	3–8	1.29–5.92	43–74	0	1.3–5.9	[48]
Turtle	1–14	0.21–10.36	21–74	−	−	[30,49,50,51]
Snake	1–50	0.02–21	2–42	−	−	[30,52,53,54]
Opossum	3.9–12.6	0.55–6.3	14–50	Rare	0.5–6.3	[55,56]
Fish	30–100	0.9–10	3–10	−	−	[30]

^1^ in 10^3^/µL; ^2^ in % of all leukocytes; (− = unknown). The values of neutrophils and heterophils in 10^3^/µL were calculated with the values from the references for % of all leukocytes and the leukocyte values in 10^3^/µL, if not already available in the cited reference.

**Table 2 ijms-21-04523-t002:** Selected granule components of neutrophils/heterophils in different species.

	Human	Mouse	Rat	Horse	Cattle	Pig	Dog	Cat	Birds	Reptiles	Opossum	Fish	References
MPO ^1^	+	+	+	+	+	+	+	+	−	−	+	+/−	[1,66,99,101,102,106]
Lysozyme	+	+	+	+	−	+	+	−	+	+	(+)	(+)	[1,101,107]
AP ^2^	+	−	+	+	+	+	+	−	−	+/−	(0)	+/−	[1,101,102,108,109]
Lactoferrin	+	+	+	+	+	+	+	+	−	−	+	−	[104,110]
β-Glucuronidase	+	+	+	+	+	+	+	+	+	+	(0)	+/-	[94,101,105,111,112,113,114]
Defensin	+	−	+	−	+	(+)	(+)	(+)	+	+/−	(+)	(+)	[19,94,115,116,117,118,119,120,121,122,123]
MP ^3^	+	+	+	+	+	+	+	(0)	+	(0)	(0)	+	[94,124,125,126,127,128,129,130,131]
Elastase	+	+	+	+	+	+	+	+	+	(0)	+	+	[94,106,122,132,133,134,135,136,137,138]
BPI ^4^	+	+	+	(+)	+	+	(+)	(+)	(+)	(+)	(0)	+	[124,132,139,140,141,142,143,144,145,146]
Cathelicidin	+	+	+	+	+	+	+	+	+	+	(+)	(+)	[19,94,116,117,122,132,147,148,149]

^1^ Myeloperoxidase, ^2^ Alkaline phosphatase, ^3^ Metalloproteinases (gelatinase, leukolysin and/or collagenase), ^4^ Bactericidal permeability increasing protein, +/- present in some species, (+) sequence in genome but not stated if present in neutrophil, (0) no data found.

**Table 3 ijms-21-04523-t003:** Selected diseases in which neutrophils/heterophils are involved.

Disease	Pathogen/Mutation	Effect on Neutrophil	Clinical Signs	Affected Species	Ref.
Anaplasmosis	*Anaplasma phagocytophilum*	Restricted movement, phagocytosis, superoxide production, adherence, transmigration, apoptosis	Unspecific, fever, impaired consciousness, lameness, arthritis, organ and lymph node swelling	Dogs, humans, ruminants, horses, wildlife, rodents, etc.	[192,198,199]
Chlamydiosis	*Chlamydia trachomatis*	Paralysis: no activation, NETosis oxidative burst, reduced cell death	Often asymptomatic, conjunctivitis, genital infection, infertility	Humans	[200,223]
Leishmaniosis/Brucellosis/Chlamydiosis	*Leishmania major/Brucella abortus/Chlamydia pneumoniae*	Trojan horse: induce apoptosis to reach macrophages	Cutaneous nodules and ulcers/abortion, inflammation/pneumonia, arthritis	Humans, rodents, birds, dogs/cattle, humans, other mammals, birds/humans, horses, reptiles, amphibians, marsupials	[104,105,106,109,223]
Leukocyte adhesion deficiency	CD18	Impaired adhesion and phagocytosis, normal morphology	Recurrent infections, sepsis, impaired wound healing, severe neutrophilia, often lethal	Humans, cattle (Holstein), dogs (Irish Setter), mice	[107,108,110]
Chédiak-Higashi syndrome	LYST (lysosomal trafficking regulator)	Disturbed formation of phagolysosome, fusion of granules (giant and pink)	Infections, partial oculocutaneous albinism, hemorrhage	Humans, cattle, cats (Persian), mice (beige), arctic foxes, mink, killer whale	[1,32,100,224]
Mucopoly-saccharidosis	Enzymes in mucopolysaccharide catabolism	Normal function, accumulation of metabolic byproducts → large azurophilic granules	Bone and cartilage defects, hepatomegaly	Humans, cattle, cats, dogs	[1,225]
Chronic granulomatous disease	NADPH oxidase	Defects in respiratory burst	Recurrent severe but not fatal infections, granulomas	Humans, dogs (Doberman Pinscher)	[1,32]
MPO deficiency	Myeloperoxidase	Delayed killing	Few clinical signs, recurrent fungal infection	Humans, dogs (Gray Collie)	[32,226]
Cyclic hematopoiesis	Neutrophil elastase (human), adaptor protein complex 3 (dog)	Mis-trafficking of neutrophil elastase	Severe cyclic neutropenia, bleeding, recurrent infections, coat color dilution, amyloidosis, often lethal	Humans, dogs	[224,227]
Pelger–Huët anomaly	Lamin B receptor	Hypolobulated nucleus	Heterozygote no signs, homozygote: skeletal deformation, susceptible to infections, usually lethal	Humans, horses, dogs, cats, rabbits, mice	[81,92]

**Table 4 ijms-21-04523-t004:** Human neutrophil cell lines.

	HL-60	NB4	PLB-985
Donor	36-year-old woman	23-year-old woman	Subclone of HL-60, with some differences in gene expression
Disease	Acute myeloblastic leukemia (AML-M2)	Acute promyelocytic leukemia in second relapse (AML-M3)	Acute myeloblastic leukemia (AML-M2)
Tissue origin	Peripheral blood	Bone marrow	HL-60
Cell type	Myeloblast	Promyelocyte	Myeloblast
Inducible cell types	Monocytes, macrophages, eosinophils, neutrophils	Macrophages, neutrophils	Monocytes, granulocytes
Differentiation towards neutrophils with (e.g.)	All-trans retinoic acid, DMSO, dibutyryl cyclic adenosine monophosphate (dbcAMP), hypoxanthine, Nutridoma	All-trans retinoic acid, DMSO, dbcAMP	All-trans retinoic acid, DMSO, dbcAMP, Nutridoma
Doubling time	40 h	35–45 h	30 h
Cytogenetics	t(8;21)	t(15;17)	−
Cathelicidin expression (LL-37)	No	No	Not known
Advantages	Inducible respiratory burst, phagocytosis and NET formation (however, all less than in primary neutrophils)		Similar characteristics to HL-60, expression of secondary granules inducible
Disadvantages	No secondary granules, failure in chemotaxis, deficient expression of late neutrophil-specific genes		Similar characteristics to HL-60
Pathogen interaction (e.g., *A. phagocytophilum*)	Infection level, reduction in defense gene transcription and oxidative burst similar to PMNs	No sufficient infection	Low infection level, minimal change in defense gene transcription and oxidative burst
References	[240,242,243,244,245,247,248,249,250]	[244,245,248,249,250,251]	[243,248,250,252,253]

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
