# Peer review of "What Is the Evolutionary Fingerprint in Neutrophil Granulocytes?"

_ijms, 2020, doi:10.3390/ijms21124523_

Round 1
Reviewer 1 Report
What Is the Evolutionary Fingerprint in Neutrophil Granulocytes?
Leonie Fingerhut, Gaby Dolz and Nicole de Buhr
The authors of this review collected and systematized a large amount of data. Comparing the structure and functioning of neutrophils in different species is not an easy task, since it is necessary to generalize the results obtained with different objects, under different conditions and using different methods. The authors mainly coped with their task and the review written by them is of interest for biologists of many specialties.
There are some comments and suggestions for authors.
Line 174
Table 1. Leukocyte and neutrophil/heterophil values in different species.
In this table, the number of leukocytes is given as the number of cells in a microliter, and the number of neutrophils is presented as a percentage of the total number of leukocytes. Both indicators vary widely. Try to compare the number of neutrophils in the blood of pigs and fish, if the number of leukocytes in pigs ranges from 2-25 (x103 / µl), and in fish 30-100, while the percentage of neutrophils in the total number of leukocytes is 12-38 and 3 -10 respectively? Summing up a large amount of data without any statistical processing in the table, in my opinion, can be replaced by comments in the text.
Lines 210-212
“The about 10-20 μm 210 big [38], spherical neutrophils possess an irregularly lobulated nucleus, which lead to the term polymorphonuclear cell. “
More precisely, neutrophils have a spherical shape in suspension, and their diameter in various studies is estimated at 6-8 mm. After attachment to solid substrates neutrophils flatten and their diameter increases to 10-20 µm or more.
Line 253
Table 2. Neutrophil/heterophil size in different species.
Quantitative data in this table (including authors own data) should be processed statistically.
Line 289
Table 3. Morphology of neutrophil/heterophils in different species.
This table describes in words the morphology of the nuclei, cytoplasm and granules of neutrophils of various species. Such terms as “pale, colorless, few eosinophil” are used to describe cytoplasm and granules. The dictionary description of the morphology of neutrophils looks archaic and does not provide the information expected from a recent scientific review. The table 3 repeats mainly the Figure 2 titled “Morphology of neutrophils and heterophils of distinct species”. Maybe instead of Table 3, the authors can make more detailed captions for photographs concerning the structure of nuclei in different species?
Line 624
Table 6 is titled “Neutrophil/heterophil values and body temperature described in different species”. However, the table does not show the number of neutrophils, but the percentage of neutrophils in relation to the total number of leukocytes, fully repeating Table 1. It is hardly worth duplicating tables. The second part of table 6, devoted to body temperature, is also not needed, as the authors write: “ one could assume a connection between physiologic ranges of body temperature and neutrophil counts in different species, but we could not find any correlation (Table 6).”[Lines 604-606].
Lines: 126-134
Chapter “Evolution”
“Whether pathogens influenced the neutrophil evolution was recently discussed in a study about NETs and malaria [35]. Neutropenia exists in the African population in the endemic areas of malaria [36] and may therefore be beneficial, as NETs have detrimental effects during malaria [35].” In this regard, it is of high interest that breeding obviously influenced NET-function. A study about the interaction of ovine NETs and helminths (Haemonchus contortus) described a different interaction of NETs from the Suffolk sheep and the St. Croix sheep. NETs of the Suffolk sheep bind the larvae less compared to NETs released by neutrophils from St. Croix sheep. How the in vivo outcome is influenced by the NET release was not finally clarified. Nevertheless, the St. Croix sheep is resistant against this parasite [37].
I cannot understand how this relates to the pathogen-induced evolution of neutrophils - there are too many assumptions.
Author Response
Answers to the reviewers ijms-834021 - first report
Dear Editors and Reviewer's,
We thank the reviewer’s for the first report and the constructive suggestions.
Please find below our answers to the comments and questions of the reviewers.
We have prepared a revised version of the manuscript, highlighting the changes from first revision using the Track Changes function in Microsoft Word, and we have prepared a point to point response in order to address the remarks out by the reviewers.
Based on the comment’s of the three reviewer's we deleted some tables and included new figures as well as one new table. Therefore, the numbers of figure and tables changed.
We thank again the reviewers for the comments to our study and tried to improve the manuscript based on the constructive comments.
With kind regards
Nicole de Buhr
Answers reviewer 1
Line 174
Table 1. Leukocyte and neutrophil/heterophil values in different species.
In this table, the number of leukocytes is given as the number of cells in a microliter, and the number of neutrophils is presented as a percentage of the total number of leukocytes. Both indicators vary widely. Try to compare the number of neutrophils in the blood of pigs and fish, if the number of leukocytes in pigs ranges from 2-25 (x103 / µl), and in fish 30-100, while the percentage of neutrophils in the total number of leukocytes is 12-38 and 3 -10 respectively? Summing up a large amount of data without any statistical processing in the table, in my opinion, can be replaced by comments in the text.
Answer: We thank the reviewer for this comment and also agree with the comment, but a statistical evaluation based on the references is not possible because more detailed information is often missing. Nevertheless, even in our own studies we have often looked for exactly such values in the literature and believe that it can help other researchers. However, we have added another column to the table and calculated the neutrophil/heterophile number in x103 / µl. Although the range here is also very wide, we think that the table is clearer compared to a text description.
Lines 210-212
“The about 10-20 μm 210 big [38], spherical neutrophils possess an irregularly lobulated nucleus, which lead to the term polymorphonuclear cell. “
More precisely, neutrophils have a spherical shape in suspension, and their diameter in various studies is estimated at 6-8 mm. After attachment to solid substrates neutrophils flatten and their diameter increases to 10-20 µm or more.
Answer: We agree with this comment and have rewritten the line 210-212 (new 215-216) and included in chapter 4.1. Size some detailed information’s (line 232 ff) about the size of neutrophils in suspension and attached neutrophils.
Line 253
Table 2. Neutrophil/heterophil size in different species.
Quantitative data in this table (including authors own data) should be processed statistically.
Answer: We thank the reviewer for this comment and processed the values in a graph (Figure 3 instead of table 2) and included several own measurements to exchange the values from references. Each measurement was conducted mainly with only one donor. We analyzed 10 neutrophils with 2 measurements per neutrophil. An ordinary one-way ANOVA was calculated followed by Tukey’s multiple comparisons test. Many significant differences were found. Nevertheless, as we only used mainly one donor (especially in the wild animals), we have not included all these significant differences and present only the p value of the ANOVA summary. All methods are described in the manuscript.
Line 289
Table 3. Morphology of neutrophil/heterophils in different species.
This table describes in words the morphology of the nuclei, cytoplasm and granules of neutrophils of various species. Such terms as “pale, colorless, few eosinophil” are used to describe cytoplasm and granules. The dictionary description of the morphology of neutrophils looks archaic and does not provide the information expected from a recent scientific review. The table 3 repeats mainly the Figure 2 titled “Morphology of neutrophils and heterophils of distinct species”. Maybe instead of Table 3, the authors can make more detailed captions for photographs concerning the structure of nuclei in different species?
Answer: Based on this constructive comment we deleted the table and included a revised version of figure 2 with a zoom (parts of cytoplasm and nuclei) and we included a new figure (figure 5) showing a zoom of neutrophils from 7 different species (each 4 neutrophils). This figure shows variations of segmentation. Furthermore, we included some sentences in chapter 4.3 and renamed it to Cytoplasm and Granules.
Line 624
Table 6 is titled “Neutrophil/heterophil values and body temperature described in different species”. However, the table does not show the number of neutrophils, but the percentage of neutrophils in relation to the total number of leukocytes, fully repeating Table 1. It is hardly worth duplicating tables. The second part of table 6, devoted to body temperature, is also not needed, as the authors write: “ one could assume a connection between physiologic ranges of body temperature and neutrophil counts in different species, but we could not find any correlation (Table 6).”[Lines 604-606].
Answer: Based on this comment we deleted table 6 to avoid repeating table 1.
Lines: 126-134
Chapter “Evolution”
“Whether pathogens influenced the neutrophil evolution was recently discussed in a study about NETs and malaria [35]. Neutropenia exists in the African population in the endemic areas of malaria [36] and may therefore be beneficial, as NETs have detrimental effects during malaria [35].” In this regard, it is of high interest that breeding obviously influenced NET-function. A study about the interaction of ovine NETs and helminths (Haemonchus contortus) described a different interaction of NETs from the Suffolk sheep and the St. Croix sheep. NETs of the Suffolk sheep bind the larvae less compared to NETs released by neutrophils from St. Croix sheep. How the in vivo outcome is influenced by the NET release was not finally clarified. Nevertheless, the St. Croix sheep is resistant against this parasite [37].
I cannot understand how this relates to the pathogen-induced evolution of neutrophils - there are too many assumptions.
Answer: We deleted line 126-134, since we know no additional findings or literature that underline these hypotheses. Perhaps there will be more reliable data about this available in the future.
Reviewer 2 Report
- In section 5, function of neutrophils need a figure to demonstrate the different fuctions, eq. transmigratoin, degranulation, ROS, NET, apoptosis.. in different species.
- In section 8, cell culture and storage, it is necessary to show a table to compare the difference of human neutrophils cells line, such as HL-60, NB4, PLB 985...
Author Response
Answers to the reviewers ijms-834021 - first report
Dear Editors and Reviewers,
We thank the reviewer’s for the first report and the constructive suggestions.
Please find below our answers to the comments and questions of the reviewers.
We have prepared a revised version of the manuscript, highlighting the changes from first revision using the Track Changes function in Microsoft Word, and we have prepared a point to point response in order to address the remarks out by the reviewers.
Based on the comment’s of the three reviewer's we deleted some tables and included new figures as well as one new table. Therefore, the numbers of figure and tables changed.
We thank again the reviewers for the comments to our study and tried to improve the manuscript based on the constructive comments.
With kind regards
Nicole de Buhr
Answers reviewer 2
Comments and Suggestions for Authors
- In section 5, function of neutrophils need a figure to demonstrate the different fuctions, eq. transmigratoin, degranulation, ROS, NET, apoptosis.. in different species.
Answer: We thank the reviewer for this suggestion and include a new figure (Figure 6) with heterophils and neutrophils and different mechanism they can undergo. We summarized the known differences as well in these mechanisms.
- In section 8, cell culture and storage, it is necessary to show a table to compare the difference of human neutrophils cells line, such as HL-60, NB4, PLB 985...
Answer: We included a table (new table 4) showing the difference of human neutrophil cell lines (HL-60, NB4 and PLB-985) in different categories based on the available data sheets and literature.
Reviewer 3 Report
This is an interesting paper covering most changes occurring in neutrophils function upon evolution.
There are some misspells and English construction phrases to review (generally speaking shorter sentences could increase the paper readability).
We believe that the section about neutrophil and cancer should be improved discussing more emerging papers in the field, e.g. for hematological malignancies:
Perez and Botta, Blood 2020
Puglisi F, JCMI, 2019
Romano A, Scientific Reports, 2020
Condamine, Sci.Immunol, 2016
and their contribution to the expansion of unconventional T-cells in sarcomas: Ponzetta, Cell 2019
These papers we are suggesting are crucial to understanding why neutrophil derangements can sustain cancer progression and should be more investigated to improve immunotherapy, as recently reviewed in Masucci, Frontiers in Onocology, 2019
Author Response
Answers to the reviewers ijms-834021 - first report
Dear Editors and Reviewers,
We thank the reviewer’s for the first report and the constructive suggestions.
Please find below our answers to the comments and questions of the reviewers.
We have prepared a revised version of the manuscript, highlighting the changes from first revision using the Track Changes function in Microsoft Word, and we have prepared a point to point response in order to address the remarks out by the reviewers.
Based on the comment’s of the three reviewer's we deleted some tables and included new figures as well as one new table. Therefore, the numbers of figure and tables changed.
We thank again the reviewers for the comments to our study and tried to improve the manuscript based on the constructive comments.
With kind regards
Nicole de Buhr
Answers reviewer 3
There are some misspells and English construction phrases to review (generally speaking shorter sentences could increase the paper readability).
Answer: We thank the reviewer for this comment and have carefully proofread the paper. Therefore, we have divided several sentences into two sentences to make them shorter and corrected misspelling.
We believe that the section about neutrophil and cancer should be improved discussing more emerging papers in the field, e.g. for hematological malignancies:
Perez and Botta, Blood 2020
Puglisi F, JCMI, 2019
Romano A, Scientific Reports, 2020
Condamine, Sci.Immunol, 2016
and their contribution to the expansion of unconventional T-cells in sarcomas: Ponzetta, Cell 2019
These papers we are suggesting are crucial to understanding why neutrophil derangements can sustain cancer progression and should be more investigated to improve immunotherapy, as recently reviewed in Masucci, Frontiers in Onocology, 2019
Answer: We thank the reviewer for this suggestion and the given references. Based on this we included in chapter 6 a new paragraph about neutrophils and cancer (lines 636 ff.)
Round 2
Reviewer 1 Report
The review under discussion is called: What Is the Evolutionary Fingerprint in Neutrophil Granulocytes?
The tables and figures provided should illustrate this topic.
- So table 1 is given in order to compare the number of neutrophils in animals, fish and birds. As I already wrote, the data given in the table do not mean anything without serious statistical processing. The range of differences in the numbers given for the same species is too wide. In addition, the authors presented the number of neutrophils in the blood of various species in fractional numbers, as in a school joke.
- In the text after the Figure 1 immediately follows Figure 3. Figure 2 is not mentioned. Signature to the Figure 3: “Neutrophil / heterophil size in different species was measured in blood smears of at least one blood donor in 10 neutrophils/ heterophils”. If I understand correctly, the authors compare the data received from a single representative of the species! It is unacceptable. To compare the sizes of neutrophils in different species, it is necessary to collect a sufficient number of individuals of each species, necessary for normal statistical processing.
- The newly added Figure 5 fully repeats Figure 2. Images of the nucleus of each species are presented 4 times in figure 5 and one more time in Figure 2. Why?
- The newly added by the authors table 4, devoted to cell lines of human neutrophils, and part of the text relating to the role of neutrophils in the development of cancer, do not support the topic indicated in the title of the review.
Author Response
- So table 1 is given in order to compare the number of neutrophils in animals, fish and birds. As I already wrote, the data given in the table do not mean anything without serious statistical processing. The range of differences in the numbers given for the same species is too wide. In addition, the authors presented the number of neutrophils in the blood of various species in fractional numbers, as in a school joke.
Answer: We thank the reviewer for this comment. Nevertheless, we think that it is important to collect the wide ranges of neutrophil numbers found for different species. This represents how wide the numbers can differ, even in a healthy state. There are no fractional numbers, as the amounts are given in 103 cells per µL (1000 cells/µl), please see line 180.
- In the text after the Figure 1 immediately follows Figure 3. Figure 2 is not mentioned. Signature to the Figure 3: “Neutrophil / heterophil size in different species was measured in blood smears of at least one blood donor in 10 neutrophils/ heterophils”. If I understand correctly, the authors compare the data received from a single representative of the species! It is unacceptable. To compare the sizes of neutrophils in different species, it is necessary to collect a sufficient number of individuals of each species, necessary for normal statistical processing.
Answer: Figure 2 is mentioned in line 234, while Figure 3 is first mentioned in line 238. Regarding figure 3: yes, we agree with the reviewers comment. For the domestic animals it would be easy to collect more donors, but as most of these values exist already in the literature we excluded now this values from these animals from our new graph. Our main problem regarding the values found in the references was that no numbers of donors was available. Based on the first reviewer round we changed therefore the table to this graph with the same amount of measured neutrophils per species. Based on the reviewer and editor comment in round 2, Figure 3 was again changed so that it now only includes the wild animals. In these species, only single individuals could be compared in our hands as no animals were especially captured for this reason. Only found wild animals that came to the veterinary clinic for another reason or to a Rescue Center in Costa Rica could be investigated and the blood was taken after the animals fully recovered. Thus, the statistical processing was deleted. The figure now only gives an impression of possible variations of neutrophil sizes. We included some informations from the literature (in first version inside table 2) to the figure legend.
- The newly added Figure 5 fully repeats Figure 2. Images of the nucleus of each species are presented 4 times in figure 5 and one more time in Figure 2. Why?
Answer: Figure 2 shows an overview of different morphologies between species including the cytoplasm. Therefore, we deleted the old table 3 and included some parts in the text as asked by the reviewer in round 1. Meanwhile, Figure 5 aims to underline how different the core shapes can be even within one species, which is why four images of one animal per selected species are shown. One example is the mouse: here the nuclei is described as ring or 8-form, both forms are presented with this pictures. Out from our own available pictures we selected human and two domestic animals (horse and cow as mammalian), mouse (as laboratory animal), opossum (as it is havening some specialities in neutrophils), turtle (reptile) and pauraque (bird). We hope that our explanation explains why we included both figure.
- The newly added by the authors table 4, devoted to cell lines of human neutrophils, and part of the text relating to the role of neutrophils in the development of cancer, do not support the topic indicated in the title of the review.
Answer: We found it important to include these topics, as variations between cell lines of different species and the feasibility of their use to replace fresh blood derived neutrophils is an important question for many researchers in the field. The new table 4 and the text about cancer was added, as it was requested by another reviewer. Because the main focus including for example the new figure 6 is to compare neutrophils from different species, we think the title fits to the main content.
Reviewer 2 Report
moderate English editing required
Author Response
moderate English editing required
Answer:
We thank the reviewer for this comment and have carefully proofread and corrected the manuscript.